# Association Between Dietary Intake and Blood Concentrations of One-Carbon-Metabolism-Related Nutrients in European Prospective Investigation into Cancer and Nutrition

**DOI:** 10.3390/nu17121970

**Published:** 2025-06-10

**Authors:** Jin Young Park, Heleen Van Puyvelde, Lea Regazzetti, Joanna L. Clasen, Alicia K. Heath, Simone Eussen, Per Magne Ueland, Mattias Johansson, Carine Biessy, Raul Zamora-Ros, José María Huerta, Maria-Jose Sánchez, Marga Ocke, Matthias B. Schulze, Catarina Schiborn, Tonje Bjørndal Braaten, Guri Skeie, Carlotta Sacerdote, Jesús Castilla, Therese Karlsson, Ingegerd Johansson, Cecilie Kyrø, Anne Tjønneland, Tammy Y. N. Tong, Verena Katzke, Rashmita Bajracharya, Cristina Lasheras, Øivind Midttun, Stein Emil Vollset, Paolo Vineis, Giovanna Masala, Pilar Amiano, Rosario Tumino, Ivan Baldassari, Elisabete Weiderpass, Elio Riboli, Marc J. Gunter, Heinz Freisling, Sabina Rinaldi, David C. Muller, Inge Huybrechts, Pietro Ferrari

**Affiliations:** 1International Agency for Research on Cancer, 69366 Lyon, France; parkjy@iarc.who.int (J.Y.P.); heleenvanpuyvelde@gmail.com (H.V.P.); learegazzetti@hotmail.fr (L.R.); biessyc@iarc.who.int (C.B.); weiderpasse@iarc.who.int (E.W.); m.gunter@imperial.ac.uk (M.J.G.); freislingh@iarc.who.int (H.F.); rinaldis@iarc.who.int (S.R.); huybrechtsi@iarc.who.int (I.H.); 2Department of Public Health and Primary Care, Faculty of Medicine and Health Sciences, Ghent University, 9000 Ghent, Belgium; e.riboli@imperial.ac.uk; 3School of Public Health, Imperial College London, London W12 0BZ, UK; j.clasen18@imperial.ac.uk (J.L.C.); a.heath@imperial.ac.uk (A.K.H.); p.vineis@imperial.ac.uk (P.V.); david.muller@imperial.ac.uk (D.C.M.); 4Health Informatics Institute, University of South Florida, Tampa, FL 33612, USA; 5Department of Epidemiology, Maastricht University, 6229 HA Maastricht, The Netherlands; simone.eussen@maastrichtuniversity.nl; 6Cardiovascular Research Institute Maastricht (CARIM), Maastricht University, 6211 LK Maastricht, The Netherlands; 7Care and Public Health Research Institute (CAPHRI), Maastricht University, 6202 AZ Maastricht, The Netherlands; 8Bevital A/S, 5608 Bergen, Norway; per.ueland@uib.no (P.M.U.); bjorn.midttun@uib.no (Ø.M.); 9Unit of Nutrition and Cancer, Cancer Epidemiology Research Programme, Catalan Institute of Oncology, Bellvitge Biomedical Research Institute (IDIBELL), 08908 L’Hospitalet de Llobregat, Spain; rzamora@idibell.cat; 10Department of Epidemiology, Murcia Regional Health Council, IMIB-Arrixaca, 30008 Murcia, Spain; jmhuerta.carm@gmail.com; 11CIBER Epidemiología y Salud Pública (CIBERESP), 28029 Madrid, Spain; mariajose.sanchez.easp@juntadeandalucia.es (M.-J.S.); jcastilc@navarra.es (J.C.); 12Escuela Andaluza de Salud Pública (EASP), 18011 Granada, Spain; 13Instituto de Investigación Biosanitaria (ibs. GRANADA), 18012 Granada, Spain; 14Department of Preventive Medicine and Public Health, University of Granada, 18071 Granada, Spain; 15National Institute for Public Health and the Environment, 3721 MA Bilthoven, The Netherlands; marga.ocke@rivm.nl; 16Department of Molecular Epidemiology, German Institute of Human Nutrition Potsdam-Rehbruecke, 14558 Nuthetal, Germany; mschulze@mail.dife.de (M.B.S.);; 17Institute of Nutritional Science, University of Potsdam, 14558 Nuthetal, Germany; 18Department of Community Medicine, UiT The Arctic University of Norway, 9037 Tromsø, Norway; tonje.braaten@uit.no (T.B.B.); guri.skeie@uit.no (G.S.); 19Unit of Cancer Epidemiology, Città della Salute e della Scienza University-Hospital, 10126 Turin, Italy; carlotta.sacerdote@cpo.it; 20Navarra Public Health Institute, 31003 Pamplona, Spain; 21Navarra Institute for Health Research (IdiSNA), 31008 Pamplona, Spain; 22Department of Internal Medicine and Clinical Nutrition, University of Gothenburg, SE 40530 Gothenburg, Sweden; therese.karlsson@gu.se; 23Department of Life Sciences, Chalmers University of Technology, SE 41296 Gothenburg, Sweden; 24Department of Odontology, Umeå University, 901 87 Umeå, Sweden; ingegerd.johansson@umu.se; 25Danish Cancer Society Research Center, Danish Cancer Society, 2100 Copenhagen, Denmark; ceciliek@cancer.dk (C.K.); annet@cancer.dk (A.T.); 26Cancer Epidemiology Unit, Nuffield Department of Population Health, University of Oxford, Oxford OX3 7LF, UK; tammy.tong@ndph.ox.ac.uk; 27Department of Cancer Epidemiology, German Cancer Research Center (DKFZ), 69120 Heidelberg, Germany; v.katzke@dkfz-heidelberg.de (V.K.); rashmita.bajracharya@dkfz-heidelberg.de (R.B.); 28Functional Biology Department, University of Oviedo, 33003 Oviedo, Spain; lasheras@uniovi.es; 29Department of Health Metrics Sciences, Institute for Health Metrics and Evaluation, University of Washington School of Medicine, Seattle, WA 98195, USA; vollset@uw.edu; 30Institute for Cancer Research, Prevention and Clinical Network (ISPRO), 50141 Florence, Italy; g.masala@ispro.toscana.it; 31Ministry of Health of the Basque Government, Sub Directorate for Public Health and Addictions of Gipuzkoa, 20013 San Sebastian, Spain; maripi.amiano@gmail.com; 32Epidemiology of Chronic and Communicable Diseases Group, Biodonostia Health Research Institute, 20014 San Sebastian, Spain; 33Spanish Consortium for Research on Epidemiology and Public Health (CIBERESP), Instituto de Salud Carlos III, 28029 Madrid, Spain; 34Hyblean Association for Epidemiological Research (AIRE-ONLUS), 97100 Ragusa, Italy; rtuminomail@gmail.com; 35Department of Epidemiology and Data Science, Fondazione IRCCS Istituto Nazionale dei Tumori, 20133 Milano, Italy; ivan.baldassari@istitutotumori.mi.it

**Keywords:** blood B-vitamin concentrations, dietary B-vitamins, one-carbon metabolism, nutrients, partial least square path modeling, principal component analysis, EPIC

## Abstract

**Background/Objectives:** We examined the association between dietary intake and blood concentrations of one-carbon metabolism (OCM)-related nutrients in the European Prospective Investigation into Cancer and Nutrition (EPIC). **Methods:** Blood concentrations and dietary intake of the vitamins riboflavin (B2), Pyridoxal 5′-phosphate (PLP and B6), folate (B9), B12, and methionine, concentrations of homocysteine, and dietary intake of betaine, choline, and cysteine were pooled from 16,267 participants in nine EPIC nested case–control studies. Correlation analyses between dietary intakes and blood concentrations were carried out. Principal component (PC) analysis identified latent factors in the two sets of measurements. **Results:** Pearson correlations between dietary intakes and blood concentrations ranged from 0.08 for methionine to 0.12 for vitamin B2, 0.15 for vitamin B12, 0.17 for vitamin B6, and 0.19 for folate. Individual dietary intakes showed higher correlations (ranging from −0.14 to 0.82) compared to individual blood concentrations (from −0.31 to 0.29). Correlations did not vary by smoking status, case–control status, or vitamin supplement use. The first PC of dietary intakes was mostly associated with methionine, vitamin B12, cysteine, and choline, while the first PC of blood concentrations was associated with folate and vitamin B6. **Conclusions:** Within this large European study, we found weak to moderate associations between dietary intakes and concentrations of OCM-related nutrients.

## 1. Introduction

One-carbon metabolism (OCM) is crucial for the synthesis and methylation of DNA, an epigenetic mechanism to regulate gene expression [1]. Attention has been paid to the key role of folate, also referred to as vitamin B9, in this metabolic network, which produces S-adenosylmethionine as the universal methyl group donor for most biological methylation reactions. However, other water-soluble B-vitamins, including riboflavin (vitamin B2), pyridoxal 5′-phosphate (PLP and vitamin B6), and cobalamin (vitamin B12), nutrients such as choline and betaine, and the amino acids methionine, cysteine, and homocysteine, are also involved in OCM [1,2,3]. An altered status of these nutrients can contribute to disruption of the OCM pathway and, therefore, interfere with DNA replication and DNA repair, affect normal gene expression patterns or alter genome stability, and ultimately might be related to the development of various cancers [4].

An accurate assessment of B-vitamin status and that of related nutrients is important in investigating the role of OCM in disease risk [5]. Self-reported dietary intake measured through food frequency questionnaires (FFQs), diet diaries, or 24 h dietary recalls (24-HDR) is prone to measurement errors. Nutritional biomarkers measured in biological specimens (e.g., plasma or serum B-vitamins and erythrocyte folate) have become an additional method to objectively assess the dietary intake of specific nutrients [6]. An additional advantage of using biomarker measurements is that they also reflect inter-individual variation due to differences in absorption, metabolism, and differential exposures to lifestyle (e.g., smoking) and dietary (e.g., nutrient–nutrient interaction) factors [5,7].

There have been few attempts to compare self-reported dietary estimates of OCM nutrients, primarily folate and vitamin B12, with blood concentrations, and the majority of these studies have aimed to validate dietary intakes [8,9,10,11,12]. In the context of folate, validation studies tend to be limited in size and show heterogeneous correlation coefficients in the range from 0.05 to 0.54 [9]. Validity is mostly inferred from correlation coefficients between the assessment instrument, such as FFQs, 24-HDR, or food records, and biomarker concentrations [5,6]. Comparisons of validation studies are challenging, even within Europe, due to different study populations, diverse biological samples, and varying analytical methodologies employed [9]. Importantly, B-vitamins and other nutrients related to OCM are challenging factors to investigate, mainly because their bio-availability and activity are not only dependent on their intake levels, but also depend on their complex mutual interactions and on their interactions with genetic factors and other dietary components [13,14].

Past studies have primarily focused on comparisons between the blood concentrations and dietary intakes of single nutrients, with a focus on folate intake [9,15]. Newer statistical approaches to identify biological patterns integrating complex interrelationships may be a complementary strategy to better understand the complex interrelationships among the different nutrients involved in OCM.

In this study, data from nine nested case–control studies within the European Prospective Investigation into Cancer and Nutrition (EPIC) were pooled to evaluate the associations between the blood concentrations and dietary intakes of B-vitamins and other OCM-related nutrients. Patterns of associations were described in univariate and multivariate analyses to elucidate the complex interrelationships among OCM variables across populations involved in a large European multicenter study.

## 2. Materials and Methods

*Study population.* The rationale and methods of the EPIC study have been previously described in detail [16,17,18]. In short, EPIC consists of 23 sub-cohorts across 10 European countries, with a wide range of cancer rates, lifestyles, and dietary habits. Between 1992 and 2000, country-specific dietary and lifestyle questionnaires were completed and blood samples were collected [16]. The present cross-sectional study used pooled data from nine existing nested matched case–control studies within EPIC on stomach [19] (*n* = 797), colorectal [20] (*n* = 3016), lung [21] (*n* = 2209), pancreatic [22] (*n* = 821), breast [23] (*n* = 4928), kidney [24] (*n* = 1055), upper aerodigestive tract [25] (UADT, *n* = 1533), and prostate cancer [26,27] (*n* = 911 and *n* = 997 in each project), as described in Table 1. The samples used in this study were from France, Italy, Spain, the United Kingdom, the Netherlands, Germany, Sweden, Denmark, and Norway. Samples from Oxford were subject to different shipping conditions (*n* = 888), and a sensitivity analysis was conducted excluding these samples. Cases and controls were matched for sex, year of birth, study center, and date of blood collection (all studies), and additionally for time of blood collection and fasting status in the pancreatic cancer study. As outlined in Appendix A, in this study, blood concentrations of riboflavin, PLP, folate, cobalamin, methionine, and homocysteine were examined, together with dietary intakes of riboflavin, vitamin B6, folate, cobalamin, methionine, cysteine, betaine, and choline in 16,267 participants. All blood concentrations were measured in all studies, except for in the study on breast cancer [23] and the first study on prostate cancer, where only folate and cobalamin were available, and the second prostate cancer study, where only riboflavin, PLP, folate, and cobalamin were available [26,27] (Table 1).

*Blood sampling and laboratory analyses.* In each of the recruitment centers, fasting or non-fasting blood samples of at least 30 mL were drawn at baseline and stored at 5–10 °C, protected from light, and transported to local laboratories for processing and aliquoting according to a standardized protocol [16,17,28]. In all countries, except Denmark and Sweden, which joined EPIC at a later stage, blood was separated into 0.5 mL aliquots (serum, plasma, red cells, and buffy coat for DNA extraction) and stored in plastic CBS straws^TM^, which were subsequently heat sealed and stored in liquid nitrogen (−196 °C). Half of the aliquots were stored at the local study center and the other half were stored in the central EPIC biorepository at the International Agency for Research on Cancer (IARC; Lyon, France). In Denmark, aliquots of 1.0 mL were stored locally at −150 °C under nitrogen vapor. In Sweden, samples were stored at −80 °C.

Laboratory analyses of different B-vitamin species have previously been described in detail [28]. All biochemical analyses were carried out in the laboratories of Bevital AS (www.bevital.no). This study included B-vitamins, i.e., riboflavin (vitamin B2), pyridoxal 5′-phosphate (PLP, vitamin B6), folate (vitamin B9), and cobalamin (vitamin B12), as well as amino acids, i.e., total homocysteine (tHcy) and methionine. The concentration levels of riboflavin (B2) and PLP (B6) were determined by LC-MS/MS [29,30], and amino acid concentrations were determined by GC-MS/MS based on methylchloroformate derivisation [31,32]. Cobalamin (B12) was determined with a *Lactobacillus leichmannii* microbiological method [33] and plasma folate was determined with a *Lactobacillus casei* microbiological method, both adapted to a microtiter plate format, and analysis was carried out on a robotic workstation (Micro-lab AT plus 2; Hamilton Bonaduz AG, Bonaluz, Switzerland) [34]. The within- and between-day coefficients of variation were 3–20% for folate [34], 3.1–13.2% for riboflavin [29], 2.6–7.4% for PLP [29], and <5% for cobalamin and tHcy [32,33], and <5% for methionine [31,32].

*Lifestyle and dietary assessment.* Lifestyle factors, including anthropometry, smoking history, physical activity, educational level, and dietary supplement use, were collected at baseline through standardized lifestyle questionnaires [16,35,36,37,38]. Daily energy intake and alcohol intake were estimated at enrolment using validated country or center-specific FFQs, designed to capture geography-specific diet at the individual level. Nearly all countries used self-administered FFQs, except in Spain and Italy (Naples and Ragusa), where questionnaires were administered by interviewers [39].

The intake of nutrients, including folate, choline, betaine, cysteine, and methionine, was estimated using the methyl group donors database (MGDB), while vitamins B2, B6, and B12 were estimated from the USDA National Nutrient Database for Standard Reference [40]. A new MGDB was compiled by matching the dietary assessment data of the EPIC cohort to four food composition databases (FCDBs), i.e., the U.S. FCDB, Canadian FCDB, German FCDB, and Danish FCDB, using standardized operating procedures. Folate intake was available in the EPIC nutrient database (ENDB) and was used for quality control [41]. A strong correlation (r = 0.81) was shown between the calculated dietary folate intakes of the new MGDB and the earlier ENDB, supporting good matching with the FCDB data [40].

The most chemically stable folate form is synthetic folic acid, which is not costly to produce and, therefore, is used for dietary supplements and food fortification. The bioavailability of food folate is commonly estimated at 50% of folic acid bioavailability [42]. To consider differences in bioavailability between food folate and folic acid used for fortification, we also assessed folate intake as dietary folate equivalents (DFEs).

*Statistical analysis*. Blood concentrations and dietary intakes were log-transformed to improve symmetry and approximate the normality of distributions, by using the log_e_(x + 1) function to avoid negative values. Blood concentrations that were recorded as being below the limit of detection were excluded from the analysis (*n* = 17). To account for missing values in blood concentrations and dietary intakes, as described in Appendix A, a multiple imputation algorithm was implemented to impute missing log-transformed values [43]. Missing values were imputed with the Multiple Imputation by Chained Equations (MICE) method in the R package ‘mice’ with a burn-in of 20 iterations. The imputation model included all OCM-related nutrient variables, as well as all adjustment factors, as detailed in the following paragraph. As the main objective of the study was to examine the main sources of variation and describe the correlation patterns of OCM-related nutrients rather than performing statistical inference, only the first imputed dataset was retained for statistical analyses.

To identify major sources of variability in concentration and dietary OCM nutrient measurements, each concentration and dietary variable was, in turn, regressed on a predefined list of covariates, including age at recruitment (continuous), alcohol intake (continuous), energy intake (continuous), BMI (continuous), case–control indicator (dichotomous), sex (dichotomous), country (categorical), smoking status (never, former, or current), and study (categorical). Models for blood concentrations also included the corresponding dietary variable (continuous) and the laboratory batch (categorical) modeled as a nested effect within the study. Partial-R^2^ statistics, expressed as percentages, were computed for each covariate as the amount of variability explained, conditional on all other covariates in the model.

To control for variability attributed to specific factors, residuals of concentration and dietary OCM-related nutrients were computed [44]. Each concentration variable was linearly regressed on country, sex, study (categorical), batch (modeled with random effects nested within study), and case–control status (categorical), while dietary variables were regressed on country, sex (categorical), case–control status (categorical), and energy intake (continuous), as displayed in Table 2 and Table 3.

Pearson correlation coefficients were computed for residual values of concentration and dietary variables. Heatmap plots were generated overall and according to smoking status (never, former, or current smokers), educational level (‘None/Primary’, ‘Secondary/Technical/Professional’, or ‘University degree’), and physical activity (‘Inactive/moderately inactive’ versus ‘Moderately active/active’). Specific heatmaps were produced according to geographic region (Southern Europe: France, Italy, and Spain; Middle Europe: UK, the Netherlands, and Germany; and Northern Europe: Denmark, Sweden, and Norway) and case–control status using residuals computed in models that did not include, in turn, country or case–control status, in order to avoid over-adjustment.

For folate measurements, a sensitivity analysis was performed comparing the correlations between concentration levels and total dietary folate (as µg/day) intakes versus DFE (µg of DFE/day). In addition, principal component (PC) analyses of, in turn, concentration and dietary residuals, were performed using the correlation matrices. Upon visual inspection of the scree plots, three PCs were retained. Last, a partial least square-path modeling (PLS-PM) analysis was performed [45], where a latent factor of the six concentration variables was identified and linearly regressed on the latent factor of the eight dietary variables, with the association estimated.

Statistical tests were two-sided, and *p*-values lower than 5% were considered to be statistically significant. Analyses were carried out with the ‘nlme’, ‘mice’, ‘corrplot’, ‘ggplot2’, ‘car’, ‘rsq’, and ‘plspm’ packages in R 4.2.1 [46].

## 3. Results

Pre-diagnostic blood concentrations and dietary intakes of the OCM-related nutrients from 16,267 participants (56% women) in nine nested case–control studies were pooled in this study, including 7653 cases and 8614 matched controls (Table 1). Linear regression models and partial-R^2^ values were assessed to evaluate the extent to which variability in the blood concentrations and dietary intake of B-vitamins and OCM metabolites was explained by several covariates. As shown in Table 2 for concentration variables, sex (range of R^2^ values, %: <0.1–2.7), country (1.6–7.7), dietary intake (0.5–3.7), study (0.2–2.3), and batch effect (3.1–7.5) were the strongest predictors, despite a lower explained variability with variations across nutrients, whereas for dietary variables, country (10.1–66.6), alcohol intake (0.5–4.1), and energy intake (25.2–60.1) explained the largest amount of variability (Table 3).

Pairwise correlation coefficients between blood concentrations and corresponding dietary variables ranged from 0.06 for methionine to 0.15 for vitamin B2, 0.15 for vitamin B12, 0.18 for vitamin B6, and 0.19 for folate (Figure 1). When excluding 888 samples from Oxford, the correlation coefficients did not change.

Comparing individual dietary components, the coefficients ranged from −0.14 between betaine and cysteine to 0.82 between methionine and cysteine. Overall, individual blood concentrations showed lower correlations among them, i.e., from −0.31 between folate and homocysteine to 0.29 between PLP and folate (Figure 1). As displayed in Figure 2, the correlation values between OCM-related concentration and dietary variables were consistent by smoking status, i.e., in never smokers (*n* = 6929), former smokers (*n* = 5154), and current smokers (*n* = 3965), and were similar to values observed in the overall study population. As reported in Appendix A, the correlations between blood and dietary variables ranged from 0.07 for methionine to 0.19 for folate among supplement users, and from 0.06 for methionine to 0.20 for folate among non-users. The correlations between intake and concentration were slightly weaker for folate, PLP (B6), and riboflavin in northern Europe compared to central and southern Europe. In addition, there were positive correlations in central Europe for dietary betaine with dietary riboflavin and B6, while inverse or no correlations were found in northern and southern Europe (Appendix A). Correlation patterns were similar by case–control status, education level, and physical activity, as displayed in Appendix A, respectively. Similar correlations were shown between folate blood concentration and total dietary folate and DFE (Appendix A).

The first three blood concentration PCs explained about 65.5% of the total observed variability (Table 4). The first PC (31.8% of the total variance) was positively associated with blood folate, PLP, riboflavin, and vitamin B12 and negatively with homocysteine, with a factor loading equal to −0.42. PC2 (17.6%) was mainly driven by blood methionine and homocysteine, while PC3 (16.1%) was positively related to riboflavin and negatively related to vitamin B12 (−0.50) and methionine (−0.59). The first three dietary intake PCs of OCM-related nutrients cumulatively explained 70.7% of the total variance. The first PC explained 43.3% of the total variability, and was positively associated with dietary methionine, cysteine, vitamin B12, choline, riboflavin, and vitamin B6, listed in descending order according to their factor loadings. The second PC (16.8%) was mainly driven by folate (0.67) and betaine (0.56), while the third PC (10.7%) was positively related to betaine (0.78) and negatively related to folate (−0.44). The pairwise correlation coefficients between the first three concentration and dietary PCs ranged from −0.08 for the majority of the comparisons to 0.18 between the first PC of the concentration and dietary variables, as displayed in Figure 3. The correlation coefficient of the PLS-PM latent factors of dietary intake and blood concentration was 0.18 (Figure 4).

## 4. Discussion

Leveraging sizeable sets of data from a large European cohort with diverse dietary and lifestyle habits, the correlation patterns between the blood concentrations and dietary intakes of OCM-related nutrients were comprehensively examined with a high precision. Pre-diagnostic biomarker concentrations measured in cancer site-specific nested case–control studies and dietary data were pooled and compared by the means of pairwise correlation coefficients. Using a holistic analysis via multivariate statistical techniques by the means of PCA and PLS-PM, we identified linearly related latent factors in the following two sets of measurements: blood concentrations and dietary intake.

In this study, the correlation coefficients between dietary intakes and blood concentrations were weak to moderate. The highest pairwise correlation coefficient, 0.19, was found between dietary and blood folate, a value similar to most observations in previous studies, in which weak to moderate correlations ranging from 0.05 to 0.54 were reported [7,9,10,11,12]. In general, higher correlations were found in studies where information on food fortification and supplement use with quantitative intake data was available. This may be due to the higher bioavailability of folic acid in participants taking supplements and/or fortified foods compared to naturally occurring dietary folate [47]. In this study, no heterogeneity of correlations by dietary supplement use was observed.

In addition, the pairwise correlation values between dietary intake and blood concentrations were similar by smoking status, separately among never, former, and current smokers. Existing evidence suggests that smoking is associated with lower concentrations of several vitamins, including B-vitamins [48], whereas dietary intakes of B vitamins do not vary significantly according to smoking status [49,50]. Studies have found that compared with never smokers, current smokers tend to have lower concentrations of PLP [51], folate, and vitamin B12 and higher concentrations of homocysteine [52]. Among the reasons explaining the short-term effects of smoking on biomarker concentrations, smoking-induced oxidative stress and inflammatory changes have been suggested to trigger an increased turnover and/or breakdown of these nutrients [53]. In this study, PLP, riboflavin, vitamin B12, and folate concentration levels were, respectively, 15%, 22%, 5%, and 20% lower in current smokers than never smokers, while dietary intakes were similar between current and never smokers. Our results are in line with previous observations that smoking reduces the bioavailability of B-vitamins, especially with the suggestion of less PLP availability, decreased B6 function, and increased catabolism in current smokers compared with never smokers [51]. The correlation values were not affected by smoking habits.

Studies comparing the dietary intakes and blood concentrations of B-vitamins largely consider comparisons of single nutrients, mainly in the context of validation studies [9,54,55]. To quantitatively investigate the complex network of OCM, with several interrelated biochemical reactions [56], latent factors of dietary intakes and biomarker levels were identified with PCA. Three PCs of blood and dietary variables explained 65.5% and 70.7% of the total variability, respectively. A correlation of 0.18 was observed between the first dietary PC and the first concentration PC.

In the current study, the correlation patterns between the OCM-related biomarker concentrations and dietary variables did not vary by education or physical activity levels, while some differences across regions were noticed. Correlations between intake and concentration were slightly weaker for folate, PLP (B6), and riboflavin in northern Europe compared to the rest of Europe. In a previous EPIC analysis of control participants included in four nested case–control studies on stomach, colorectal, lung, and pancreatic cancers, a decreasing north-to-south gradient in the plasma levels of vitamins B2 and B6 and an increasing north-to-south gradient of folate (in women) and in several amino acids involved in OCM were observed [28]. The study highlighted that geographical heterogeneity in the concentrations of OCM-related biomarkers might have implications for the incidence of major chronic diseases in western Europe.

To our knowledge, this is the largest cross-sectional study that has described the level of agreement between the blood concentrations and dietary intakes of OCM-related nutrients, both as individual nutrients and using an integrated analytical approach to account for the complex interrelationships of OCM. The large sample size and availability of harmonized epidemiological lifestyle and dietary data allowed for an extensive list of informative comparisons to be evaluated. In EPIC, blood samples were processed with standardized protocols [16], stored in liquid nitrogen, and assayed in the same laboratory at Bevital in Bergen, Norway (http://www.bevital.no) [57]. These procedures possibly maintained the long-term integrity of biological samples and minimized inter-laboratory variability that can be substantial, at least for folate concentration [58]. However, the present study pooled concentration data acquired from case–control studies that took place at different points in time, potentially introducing systematic between-study variability affecting concentrations. Two analytical strategies were implemented to address this. An R-squared analysis was carried out for each concentration variable to quantify the amount of variability attributable to several key factors, including the participants’ age, BMI, sex, and country, as well as relevant design variables, such as study and batch within study. This evaluation highlighted that once the role of other factors was accounted for, variables like country (with an R-squared value ranging from 1.6% to 7.7% of the total variability explained) and batch within study (from 3.1% to 7.5%) were the strongest predictors. Interestingly, the dietary counterparts of specific concentration nutrients and smoking status explained a limited proportion of variability, with R-squared values in the range from 0.5% to 3.7% and from 0.2% to 3.1%, respectively. Then, on the basis of these observations and capitalizing on the experience we accumulated in pooling molecular data across studies [59], residual values were computed in separate linear models to control for the variability in concentrations due to design variables.

Our study had limitations. First, as blood concentrations of the OCM-related nutrients were based on a single blood sample per study participant, it was not possible to assess within-person variability. However, a study examining the over-time stability of several biomarkers found that the reproducibility of B-vitamin concentrations was sufficient with one sample collection per study participant [60]. The stability of samples in long-term storage was another factor which could have influenced concentration levels, yet EPIC samples were stored in liquid nitrogen to maintain their integrity. Second, although biomarker blood concentrations are considered as objective measures of the diet that do not suffer from recall bias or exposure misclassification [6] and provide informative comparisons with dietary intakes [7,10,61], circulating concentrations reflect endogenous processes related to their metabolism, also involving bacteria in the gastro-intestinal tract [62], as well as lifestyle factors [10] and ultimately dietary intakes. Nonetheless, the objective of the study was a comprehensive evaluation of OCM-related nutrients, describing the patterns of associations between intake and concentration values to elucidate the complex interrelationships of OCM variables in the EPIC study population, rather than a validation study of intake data. Third, within the EPIC cohort, dietary intakes were estimated using self-administered FFQs [16], which are prone to measurement errors due to study participants’ inability to accurately recall their past diet [63]. Errors may have also originated from the use of inaccurate food composition tables, although the use of country-specific food composition tables was compared in EPIC, showing good correlations between nutrient intakes derived from different tables [40,64]. Fourth, it is worthwhile to note that low correlations could reflect the inherent difference in the time frame of exposure windows covered by concentration levels versus dietary assessments. While plasma or serum concentrations reflect recent dietary exposure over the previous days to months [65], dietary intakes assessed by FFQs mostly cover habitual diet over the year preceding their administration. It is worth highlighting the importance of assessing dietary intake over time, as well as of collecting longitudinal biological samples to appreciate changes in exposure throughout the lifetime of study participants. In addition, information on the type and dose of dietary supplement used was not available and did not contribute to the estimation of study participants’ dietary nutrient intake. Nevertheless, at the time of data collection, mostly between 1990 and 2000 [16], B-vitamin-containing supplements were not the most frequently consumed types of supplements among those 35.5% of supplement users reported in the EPIC cohort [66], nor was folic acid fortification a common practice in Europe, with the exception of specific population groups, such as pregnant women. Also, the correlations observed in our study did not vary by reported supplement use. Another limitation is that many EPIC study populations were not representative of the general populations. However, under the assumption that observed correlations were linear beyond the (log-transformed) ranges of exposure variability, the correlation patterns observed in this study can still inform on the level of agreement between concentration and dietary measurements in the general population and EPIC study population. Lastly, it is important to note that approximately 30–40% of concentration data for PLP, riboflavin, methionine, and homocysteine were missing in our study, while other variables had less than 1.5% of missing data, and those missing values were imputed using a multiple imputation algorithm.

## 5. Conclusions

In summary, the correlations between OCM-related dietary intake and concentrations, as well as the correlations between corresponding data-driven blood and dietary latent factors, were weak to moderate, suggesting that dietary intake and concentration levels are not interchangeable and possibly express different quantities. Leveraging an unprecedented large dataset of existing dietary and biomarker data, this study provides relevant individual-level information on nutrients of OCM within a unique international setting of the EPIC cohort. The results of this study will inform future association studies between OCM-related nutrients and the risk of specific diseases, which might guide clinical practice and public health policy.

## Figures and Tables

**Figure 1 nutrients-17-01970-f001:**
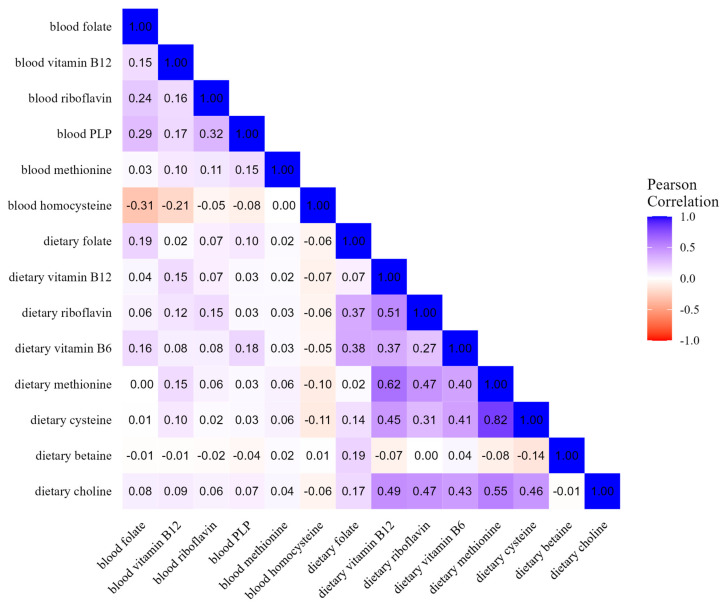
Heatmap of OCM-related blood concentrations and dietary intakes (*n* = 16,250).

**Figure 2 nutrients-17-01970-f002:**
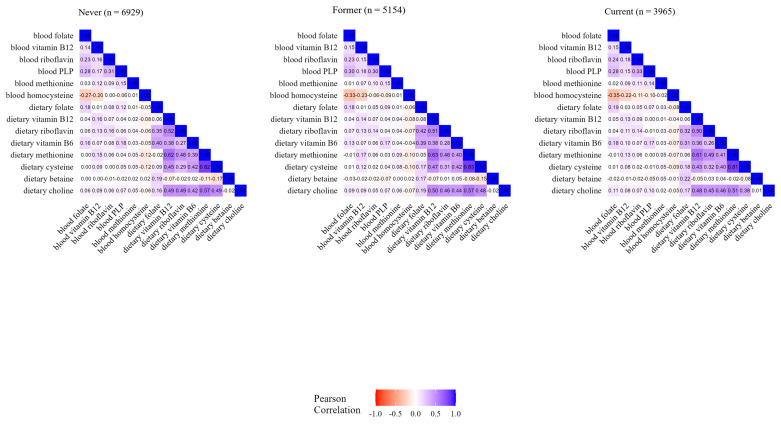
Heatmap of OCM-related blood concentrations and dietary variables by smoking status.

**Figure 3 nutrients-17-01970-f003:**
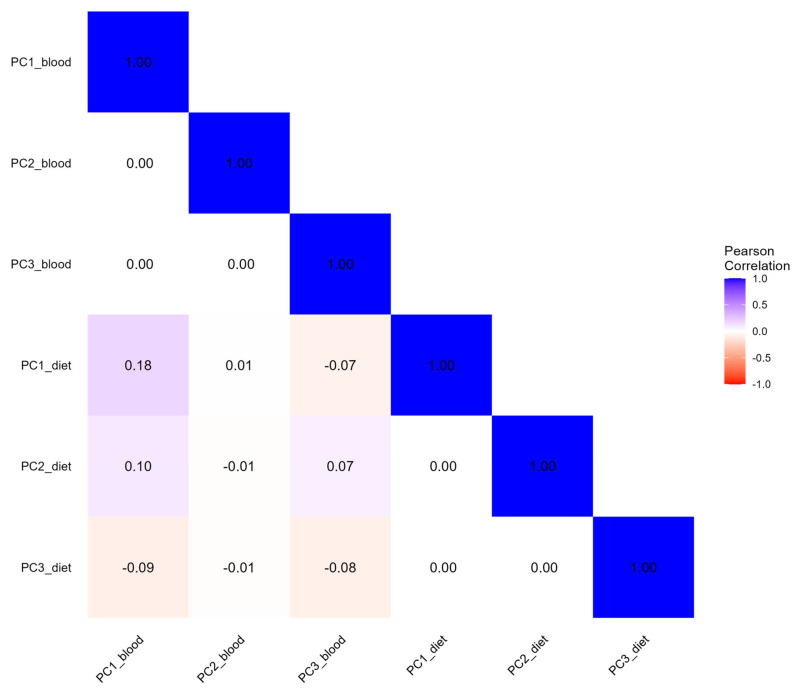
Correlation heatmap of OCM-related concentration and dietary principal components. PCs Blood indicate principal components from concentration variables. PCs Diet indicate principal components from dietary variables.

**Figure 4 nutrients-17-01970-f004:**
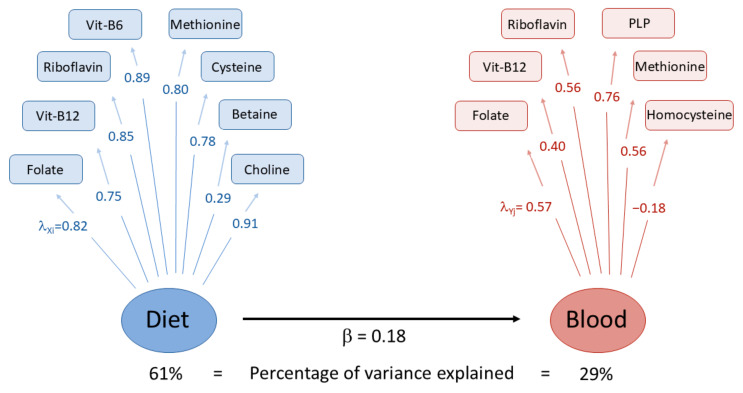
Partial least square path modeling (PLS-PM) analysis of OCM-related dietary intakes (Diet) and blood concentrations (Blood). Loading factors for dietary (λ_X_) and concentration (λ_Y_) variables indicate the correlation level between each variable and the estimated latent factors (in blue and red circles for dietary intakes and blood concentrations of nutrients, respectively). The estimated regression coefficient (β) between dietary and concentration latent factors is also reported. The estimate of β also expresses their correlation coefficient. PLP: Pyridoxal 5′-phosphate.

**Table 1 nutrients-17-01970-t001:** Characteristics of the cancer site-specific case–control studies included in the pooled analysis.

Study	Age Range	Blood Biomarkers ^1^	Number of Biomarkers	N	Study Size		Case-Control Status		Country								
					Men	Women	Cases	Controls	France	Italy	Spain	United Kingdom ^3^	The Netherlands	Germany	Sweden	Denmark	Norway
Stomach	36–75	B2, B6, B9, B12, H, M	6	797	471	326	293	504	9	189	130	85	59	132	128	65	-
Breast	26–77	B9, B12	2	4928	-	4928	2499	2429	801	1301	432	812	640	680			262
Kidney	36–75	B2, B6, B9, B12, H, M	6	1055	576	479	559	496	26	174	104	130	88	238	64	223	8
Lung	34–77	B2, B6, B9, B12, H, M	6	2209	1333	876	858	1351	3	389	382	521	329	438	147	-	-
Pancreas	30–76	B2, B6, B9, B12, H, M	6	821	395	426	449	372	18	78	73	85	71	103	225	164	4
CRC ^2^	30–77	B2, B6, B9, B12, H, M	6	3016	1384	1632	1191	1825	75	435	354	611	431	447	533	116	14
UADT ^2^	34–76	B2, B6, B9, B12, H, M	6	1533	1095	438	808	725	2	139	198	232	142	195	76	545	4
Prostate I	44–77	B9, B12	2	911	911		488	423		110	158	284	43	314	2	-	-
Prostate II	40–77	B2, B6, B9, B12,	4	997	997		508	489		157	202	325	36	277	-	-	-
			Total	16,267	7162	9105	7653	8614	934	2972	2033	3085	1839	2824	1175	1113	292

^1^ B2: riboflavin; B6: pyridoxal 5′-phosphate (PLP); B9: folate; B12: cobalamin; H: homocysteine; M: methionine. Concentrations of betaine and choline were also available in the studies on stomach, kidney, colorectal, and UADT cancers, but these measurements were not examined in this study. ^2^ CRC: colorectal cancer; UADT: upper aero-digestive tract cancers; ^3^
*n* = 443 health conscious, and *n* = 2642 general population.

**Table 2 nutrients-17-01970-t002:** Partial R^2^ values (R^2^ expressed as percentages, %) and associated *p*-value ^1^ for OCM-related blood concentration measurements in relation to an a priori defined list of determinants. Concentration values were log-transformed.

	df ^2^	Riboflavin (B2)		PLP (B6)		Folate (B9)		Cobalamin (B12)		Methionine		Homocysteine	
		R^2^ (%)	*p*-Value	R^2^ (%)	*p*-Value	R^2^ (%)	*p*-Value	R^2^ (%)	*p*-Value	Partial R^2^ (%)	*p*-Value	R^2^ (%)	*p*-Value
Age at recruitment	1	0.1	0.001	0.2	<0.001	0.2	<0.001	<0.1	0.556	0.2	<0.001	0.8	<0.001
Alcohol intake	1	<0.1	0.002	1.4	<0.001	0.5	<0.001	<0.1	0.002	<0.1	0.112	0.6	<0.001
BMI	1	<0.1	0.048	1	<0.001	<0.1	<0.001	<0.1	0.131	<0.1	0.323	<0.1	0.193
Case–control status	1	<0.1	0.247	0.4	<0.001	<0.1	<0.001	<0.1	0.646	0.2	<0.001	0.2	<0.001
Country	8	2.9	<0.001	2	<0.001	2.6	<0.001	1.6	<0.001	2.5	<0.001	7.7	<0.001
Energy intake	1	0.5	<0.001	1.8	<0.001	2	<0.001	0.6	<0.001	0.1	0.003	0.3	<0.001
Dietary intake ^3^	1	1.5	<0.001	2.9	<0.001	3.7	<0.001	2.2	<0.001	0.5	<0.001	-	-
Sex	1	0.5	<0.001	<0.1	0.076	0.3	<0.001	0.3	<0.001	1.8	<0.001	2.7	<0.001
Smoking status	3	1.9	<0.001	3.1	<0.001	1.5	<0.001	0.2	<0.001	0.8	<0.001	0.7	<0.001
Study	- ^2^	1.4	<0.001	2	<0.001	0.2	<0.001	2.2	<0.001	2.3	<0.001	1.1	<0.001
Study/batch	- ^2^	7.5	<0.001	3.1	<0.001	4.3	<0.001	4.3	<0.001	3.7	<0.001	4.6	<0.001

^1^ *p*-Values were obtained from F-test on type-III sum of squares; ^2^ df: degree of freedom; for the variables ‘Study’ and ‘Batch within Study’, the df reflects the availability of samples in each study and batch, and is equal to 6 and 159 for riboflavin, 6 and 158 for PLP, 8 and 218 for folate and cobalamin, 5 and 150 for methionine, and 5 and 155 for homocysteine; ^3^ Dietary intake of the corresponding nutrient; BMI, body mass index; OCM, one-carbon metabolism; PLP, pyridoxal 5′-phosphate.

**Table 3 nutrients-17-01970-t003:** Partial R^2^ values (R^2^ expressed as percentages, %) and associated *p*-value ^1^ for OCM-related dietary intakes in relation to a predefined list of determinants. Dietary values were log-transformed.

	df ^2^	Riboflavin (B2)		PLP (B6)		Folate (B9)		Cobalamin (B12)		Methionine		Cysteine		Betaine		Choline	
		R^2^ (%)	*p*-Value	R^2^ (%)	*p*-Value	R^2^ (%)	*p*-Value	R^2^ (%)	*p*-Value	R^2^ (%)	*p*-Value	R^2^ (%)	*p*-Value	R^2^ (%)	*p*-Value	R^2^ (%)	*p*-Value
Age at recruitment	1	<0.1	0.31	<0.1	0.215	<0.1	<0.001	<0.1	0.034	<0.1	0.98	0.4	<0.001	<0.1	0.068	<0.1	0.69
Alcohol intake	1	2.8	<0.001	0.6	<0.001	1.8	<0.001	0.2	<0.001	2.5	<0.001	4.1	<0.001	0.7	<0.001	0.5	<0.001
BMI	1	0.4	<0.001	0.5	<0.001	<0.1	0.121	1.1	<0.001	1.7	<0.001	1.6	<0.001	<0.1	0.005	1.1	<0.001
Case–control status	1	<0.1	0.352	<0.1	0.217	<0.1	0.021	<0.1	0.835	<0.1	0.109	<0.1	0.302	<0.1	0.411	<0.1	0.633
Country	8	28.4	<0.001	33.8	<0.001	25.1	<0.001	10.1	<0.001	13.6	<0.001	19.2	<0.001	66.6	<0.001	26.8	<0.001
Energy intake	1	54.2	<0.001	48.5	<0.001	53.4	<0.001	25.9	<0.001	52.1	<0.001	60.1	<0.001	25.2	<0.001	57.2	<0.001
Study	8	<0.1	0.148	<0.1	0.254	<0.1	0.297	<0.1	0.612	0.1	0.007	0.2	<0.001	0.2	<0.001	0.1	0.017
Sex	1	0.6	<0.001	<0.1	0.02	0.4	<0.001	0.1	<0.001	<0.1	<0.001	<0.1	0.016	0.2	<0.001	0.1	<0.001
Smoking status	3	0.3	<0.001	0.2	<0.001	0.3	<0.001	0.3	<0.001	<0.1	0.002	<0.1	0.041	<0.1	0.212	0.2	<0.001

^1^ *p*-Values were obtained from F-test on type-III sum of squares; ^2^ df: degree of freedom; BMI, body mass index; OCM, one-carbon metabolism; PLP, pyridoxal 5′-phosphate.

**Table 4 nutrients-17-01970-t004:** Factor loadings of the first three principal components of OCM-related blood concentrations and dietary intakes.

Concentration Variable	PC1	PC2	PC3
Folate (B9)	0.50	−0.27	0.27
Cobalamin (B12)	0.37	−0.18	−0.50
Riboflavin (B2)	0.43	0.33	0.36
PLP ^1^ (B6)	0.47	0.33	0.29
Methionine	0.17	0.66	−0.59
Homocysteine	−0.42	0.50	0.32
Proportion of explained variance (%)	31.8	17.6	16.1
Cumulative explained variance (%)	31.8	49.4	65.5
**Dietary Variable**	**PC1**	**PC2**	**PC3**
Folate (B9)	0.17	0.67	−0.44
Cobalamin (B12)	0.41	−0.13	0.21
Riboflavin (B2)	0.36	0.20	0.02
PLP ^1^ (B6)	0.34	0.26	−0.32
Methionine	0.46	−0.25	0.19
Cysteine	0.42	−0.21	−0.03
Betaine	−0.04	0.56	0.78
Choline	0.40	0.01	0.10
Proportion of explained variance (%)	43.3	16.8	10.7
Cumulative explained variance (%)	43.3	60.1	70.7

^1^ PLP, pyridoxal 5′-phosphate.

## Data Availability

The original contributions presented in the study are included in the article/Appendix A, further inquiries can be directed to the corresponding author.

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
