# Peer review of "Association Between Dietary Intake and Blood Concentrations of One-Carbon-Metabolism-Related Nutrients in European Prospective Investigation into Cancer and Nutrition"

_nutrients, 2025, doi:10.3390/nu17121970_

Round 1
Reviewer 1 Report
Comments and Suggestions for Authors
The paper “Association Between Dietary Intake and Blood Concentrations of One-Carbon Metabolism-Related Nutrients in the European Prospective Investigation into Cancer and Nutrition” focused on the relationship between dietary intake and blood concentration of nutrients related to carbon metabolism in a European population, which provides an important basis for follow-up studies. Here are some specific comments.
Comments:
Q1. The correlation results of this study are similar to those of previous ones, thus the innovative presentation of this paper is suggested to be further strengthened.
Q2. Does this study's large sample size and multiple European countries affect extrapolation because it does not appear to involve the general population?
Q3. Multiple nutrients were involved in one-carbon metabolism. It is suggested to discuss the impact of complex interactions between nutrients on the outcome.
Q4. Research has found that the correlation between the intake and concentration of certain nutrients varies by region, but the reasons for these differences have not been thoroughly explored. Further stratification by region could be conducted to analyze the impact of dietary patterns, living habits, environmental factors, and other aspects in different regions on the results.
Q5. The discussion on the guiding significance of the research results for public health and clinical practice is insufficient.
Q6. The data collection period was mainly from 1992 to 2000, during which the dietary structure and lifestyle of the European population changed. Whether the dietary pattern has changed further in the past two decades, and whether the conclusions of this study apply to the present.
Reviewer 2 Report
Comments and Suggestions for Authors
This manuscript presents a comprehensive and well-conducted analysis of the association between dietary intake and blood concentrations of one-carbon metabolism (OCM)-related nutrients using data from the large-scale EPIC cohort. The study design is robust, with an impressive sample size and inclusion of multiple nested case-control studies across Europe. The use of multivariate techniques such as principal component analysis (PCA) and partial least squares-path modelling (PLS-PM) is commendable, allowing the authors to address the complexity of nutrient interactions in OCM. The manuscript is well-written, methodologically sound, and makes a significant contribution to the field of nutritional epidemiology, particularly in the validation and interpretation of dietary intake and biomarker data.
While the correlations between intake and biomarkers are modest, the interpretation could be more nuanced. Are there biological reasons (e.g., nutrient bioavailability, transport, or metabolism) that could explain these discrepancies, particularly for methionine and riboflavin?
The authors might consider elaborating on how their findings inform the use of FFQs or biomarkers in future epidemiological research.
The imputation method is briefly mentioned, but more detail on the multiple imputation process (e.g., number of imputations, variables included, diagnostics performed) would enhance transparency.
While the discussion notes the different time frames of diet and blood measures, it may be useful to further emphasize this as a key limitation that inherently affects correlation analyses.
Given the study’s reliance on dietary data without quantitative supplement information, consider discussing how this may affect interpretation, particularly in countries or subgroups where supplement use is more prevalent.
Figures 1–4: These are informative but could benefit from slightly enhanced clarity, especially regarding labeling of PCs and scales.
Table 4: Please clarify the rationale for selection of three PCs and whether they were retained based solely on the scree plot.
